# IRS-Assisted Dual-Mode Relay-Based Adaptive Transmission

**DOI:** 10.3390/s25247492

**Published:** 2025-12-09

**Authors:** Dabao Wang, Yanhong Xu, Zhangbo Gao, Hanqing Ding, Shitong Zhu, Zhao Li

**Affiliations:** 1Institute of Remote Sensing Satellite, China Academy of Space Technology, Beijing 100094, China; cast_wdb@126.com; 2School of Cyber Engineering, Xidian University, Xi’an 710126, China; yhxu_1@stu.xidian.edu.cn (Y.X.); 24151213751@stu.xidian.edu.cn (Z.G.); a649054505@163.com (S.Z.); 3School of Electronics and Information, Zhengzhou University of Light Industry, Zhengzhou 450001, China; dinghanqing@zzuli.edu.cn

**Keywords:** intelligent reflecting surface, relay, beamforming, adaptive transmission

## Abstract

To address the challenges posed by increased power consumption in traditional active relays and the difficulties associated with countering channel fading for Intelligent Reflecting Surfaces (IRSs), we propose a dual-mode relay (DMR). This relay can dynamically switch between two operational modes: active relaying and passive IRS reflection. The DMR allows its units (DMRUs) to select their operational modes based on channel conditions. This capability enables the transmission of composite-mode signals, which consist of both active relaying components and IRS-reflected components. This dynamic switching enhances adaptation to the wireless environment. Furthermore, under the constraint of limited transmit power, we introduce a DMR-based Adaptive Transmission (DMRAT) method. This approach explores all possible DMR operational modes and employs the Alternating Optimization (AO) algorithm in each mode to jointly optimize the beamforming matrices of both the transmitter and the DMR, along with the reflection coefficient matrix of the IRS. Consequently, this maximizes the data transmission rate for the target communication pair. The optimal DMR mode can then be determined based on the optimized data rate for the target communication across various operational modes. Simulation results demonstrate that the proposed method significantly enhances the data transmission rate for the target communication pair.

## 1. Introduction

The rapid advancement of wireless communication technologies has led to a significant increase in both the number of users and the traffic load within communication systems. To provide high-quality data services to mobile users, ensuring adequate network coverage is essential. Relay technology, which employs relay nodes to receive and forward signals, can extend communication range and enhance users’ signal reception quality [1]. Decode-and-Forward (DF) and Amplify-and-Forward (AF) are two of the most prevalent relaying schemes in this domain. However, these conventional active relay technologies fundamentally rely on active components, including carrier signal generators, analog-to-digital converters, and power amplifiers [2], resulting in substantial hardware complexity and significant power consumption. Intelligent Reflecting Surfaces (IRSs) consist of numerous passive reflecting elements. Each element can dynamically adjust its reflection coefficient under the control of an IRS controller. As a low-cost, flexibly deployable, and fully passive device, IRS can be applied in various domains such as physical-layer security [3,4,5,6], enhancement of wireless data transmission [7,8], and Integrated Sensing and Communication (ISAC) [9,10,11]. Therefore, IRS is regarded as a revolutionary technology in wireless communications. It offers new design paradigms for signal forwarding and relaying.

Although there has been considerable theoretical and practical research on IRS, their limited signal processing capabilities and passive nature make them more susceptible to channel fading. Moreover, practical implementations of IRS encounter constraints such as limited operating frequency bands [12] and insufficient dynamic adjustment capabilities [13]. Consequently, some research efforts have shifted toward integrated designs that combine IRS with relay techniques. The authors of [14] investigated an IRS-assisted two-way Amplify-and-Forward relaying system. In their design, one IRS is deployed between the transmitter and the relay node, while another is positioned between the relay and the receiver to enhance signal transmission. By optimizing the phase shift matrices of both IRSs, the signals forwarded by the relay and those transmitted from the source are constructively superimposed in phase at the destination receiver. This effectively reduces the bit-error rate (BER).

In [15], the authors proposed an Unmanned Aerial Vehicle (UAV)-aided communication scheme that jointly employs an active relay and an IRS. In this study, the UAV is equipped with a multi-antenna AF relay and an IRS consisting of multiple passive reflecting elements. The relay and IRS can operate both independently and collaboratively to forward signals. Through joint optimization of the UAV’s trajectory, the IRS reflection coefficients, and the relay beamforming parameters, the achievable data rate at the target receiver is maximized.

Despite the studies in [14,15] that integrate IRS with relays, they primarily treat the passive IRS as a supplementary tool to enhance the performance of active relays. They do not consider the option to flexibly replace active relays with IRS (i.e., the adaptive selection between relay and IRS). In fact, both relay and IRS have their own advantages and disadvantages regarding energy efficiency, hardware and computational complexity, and coverage. The ability to adaptively replace active relays with passive IRS based on dynamic wireless channel conditions holds significant research importance. This approach can reduce power consumption or enhance data transmission under fixed power constraints, thus improving communication performance from the source to the destination node. Therefore, this paper proposes a dual-mode relay (DMR) capable of dynamically switching between an active relaying mode and a passive IRS reflection mode. The DMR consists of multiple dual-mode relay units (DMRUs). Each DMRU can flexibly switch between active relaying and passive IRS reflection based on wireless channel conditions. Multiple DMRUs cooperatively process the incident signal in a composite manner to relay data information. Furthermore, under a transmit power constraint, we design a DMR-based Adaptive Transmission (DMRAT) scheme. This scheme evaluates all possible DMR operational modes. For each mode, it jointly optimizes the beamforming matrix at the transmitter and the operational parameters of the DMRUs. These DMRUs can operate in either active or passive mode using an Alternating Optimization (AO) algorithm, with the goal of improving the data rate at the receiver.

The rest of this paper is organized as follows. Section 2 describes the system model, while Section 3 presents the design of DMRAT. Section 4 evaluates the performance of the proposed method. Finally, we conclude this paper in Section 5.

Throughout this paper, we will use the following notations. The set of complex numbers is denoted as C, while vectors and matrices are represented by lower-case and upper-case bold letters. ${\left(\cdot \right)}^{T}$ and ${\left(\cdot \right)}^{H}$ represent matrix transpose and conjugate transpose, respectively. |·| denotes the absolute value of a complex number.

## 2. System Model

We consider a communication system consisting of one transmitter (Tx), one receiver (Rx), and a dual-mode relay (DMR), as illustrated in Figure 1. The Tx is equipped with $N_{T}$ antennas and transmits with power $P_{T}$, while the Rx has $N_{R}$ antennas. Due to obstacles or severe channel fading between the Tx and Rx, no direct link satisfying the transmission quality requirement is available [16]. The DMR consists of $N_{DM}$ DMRUs, each capable of dynamically switching between active relaying and passive IRS reflection modes for signal forwarding, under the control of a control unit (CU) co-located with the Tx. When a DMRU operates in active relaying mode, it functions as a full-duplex DF relay. The total transmit power consumed by the DMR is denoted as $P_{DM}$. Assuming that the signal processing delays for both DF relaying and passive IRS reflection are negligible, the signals actively forwarded and passively reflected by all DMRUs can arrive at the Rx simultaneously. While practical DF relaying incurs higher processing latency than IRS reflection, synchronization can be achieved by employing IRS architectures with controllable time delays [17] or using active instantaneous relay [18,19]. One can minimize the delay differences between the signal components transmitted via the IRS reflection path and the active relaying path to an acceptable level, validating the synchronization assumption made in our system.

Prior to communication, the Tx broadcasts a pilot signal with all DMRUs of the DMR operating in passive IRS reflection mode. The Rx estimates the Channel State Information (CSI) [20] between itself and both the IRS and the Tx, denoted as H and G in Figure 1. For instance, both H and G can be acquired using the tensor-based estimation scheme [21], which requires $N_{DM}\times N_{T}$ pilot symbols. Subsequently, the Rx feeds the estimated CSI back to the Tx via the DMR-assisted reflecting link [22]. The Tx then transmits the data $x={\left[x_{1}\;x_{2}\;\ldots \;x_{N_{I}}\right]}^{T}$. The Tx precodes x using the beamforming matrix $W_{Tx}\in C^{N_{T}\times N_{I}}$ and then map the pre-processed output to the $N_{T}$ transmit antennas for transmission. The channels between the Tx and the DMR, and between the DMR and the Rx, are represented by $H\in C^{N_{DM}\times N_{T}}$ and $G\in C^{N_{R}\times N_{DM}}$, respectively. We employ a spatially uncorrelated Rayleigh flat fading channel model to characterize these channels, where the elements are independent and identically distributed complex Gaussian random variables with zero mean and unit variance. Both H and G exhibit quasi-static flat fading [23] characteristics.

## 3. Design of DMRAT

This section presents the design of the DMRAT under the constraint of the total transmit power at both the Tx and the DMR. For simplicity, we assume $N_{T}=N_{R}=N_{DM}=N_{I}=M$, where *M* is a positive integer.

### 3.1. Basic Signal Processing in the DMR

In practical applications, when the channel quality from the Tx to the Rx via the IRS is sufficiently good, the DF relay may not be necessary. In such cases, using only IRS reflection can provide the Rx with adequate signal reception quality. Conversely, when the channel condition is poor, using IRS reflection alone may not be sufficient to ensure good reception quality at the Rx. Therefore, it is necessary to set some DRMUs to active relaying mode to compensate for channel fading, albeit at the cost of some transmit power. To reduce the power consumption of the DMR, the Tx can configure a subset of the DMRUs to operate in IRS mode. As a result, some DMRUs operate in DF relaying mode, while others function in passive IRS reflection mode. The Rx then receives a composite signal constructed from multiple signal components originating from the DMRUs operating in these two distinct modes. To control the operating modes of the *M* DMRUs in the DMR, we define a mode selection matrix $S=\mathrm{diag}(s_{1},s_{2},\ldots ,s_{M})$, where each element $s_{m}\in \left\{0,1\right\}$ (m∈{1,2,…,M}) indicates whether the *m*-th DMRU operates in DF relaying mode or IRS reflection mode.

Let $U_{m}$ denote the *m*-th DMRU. We define $\Omega_{DF}$ as the set of DMRUs operating in DF relaying mode and $\Omega_{IRS}$ as the set of DMRUs operating in IRS reflection mode. $\Omega_{DF}\cup \Omega_{IRS}=\Omega$ holds, where Ω represents the entire set of DMRUs. For each $U_{m}\in \Omega_{DF}$, the mode indicator is set as $s_{m}=1$; conversely, for $U_{m}\in \Omega_{IRS}$, $s_{m}=0$. Under the control of the mode selection matrix S, the received signal at the Rx can be expressed as:(1)$$y_{DM}=\sqrt{P_{DM}}GSW_{DF}x+\sqrt{P_{T}}G\left(S⊕I\right)\Phi HW_{Tx}x+n.$$

The first term on the right-hand side (RHS) of Equation (1) represents the signal forwarded by the DMRUs operating in the DF relaying mode, while the second term corresponds to the signal reflected by the DMRUs operating in the IRS reflection mode. $W_{DF}=\left[w_{DF,1}\;w_{DF,2}\;\ldots \;w_{DF,M}\right]$ denotes the beamforming matrix for the DMRUs in $\Omega_{DF}$, where $w_{DF,m}\in C^{M\times 1}$ is the *m*-th column vector of $W_{DF}$. n represents the Additive White Gaussian Noise (AWGN) vector, whose elements have zero mean and variance $\sigma_{n}^{2}$. The reflection coefficient matrix of the DMRUs in $\Omega_{IRS}$ is represented as $\Phi =\mathrm{diag}(\beta_{1}e^{j\alpha_{1}},\ldots ,\beta_{m}e^{j\alpha_{m}},\ldots ,\beta_{M}e^{j\alpha_{M}})$, where $\alpha_{m}$ and $\beta_{m}$ denote the phase coefficient and amplitude coefficient, respectively, of the *m*-th DMRU operating in IRS mode. $W_{Tx}=\left[w_{Tx,1}\;w_{Tx,2}\;\ldots \;w_{Tx,M}\right]$ represents the beamforming matrix at the Tx. The matrices S and S⊕I (where I is the identity matrix and ⊕ denotes element-wise modulo-2 addition) are applied to $W_{DF}$ and Φ, respectively, to select the operating mode of each DMRU. When $s_{m}=0$, the matrix S sets the *m*-th row elements of $W_{DF}$ to zero; when $s_{m}=1$, S⊕I sets the *m*-th reflection coefficient in Φ to zero, i.e., $\beta_{m}e^{j\alpha_{m}}=0$.

Under the total transmit power constraint $P_{total}=P_{T}+P_{DM}$, we let $P_{T}=P_{DM}=P_{total}/2$ for example. The subsequent analysis can be applied to other power allocations. The Rx employs a filtering matrix $F=[f_{1}\;\ldots f_{m}\;\ldots f_{M}]$ for post-processing the received signal $y_{DM}$, where $f_{m}\in C^{M\times 1}$ is the filter vector corresponding to the desired data $x_{m}$. The estimated signal can be expressed as ${\hat{y}}_{DM}=F^{H}y_{DM}$. The data rate at the Rx is calculated as [24]:(2)$$R_{DM}=\sum_{m=1}^{M}\log_{2}\left\{1+\frac{P_{total}{\left|f_{m}^{H}\left(h_{eq,m}^{DF}w_{DF,m}+h_{eq,m}^{IRS}w_{Tx,m}\right)\right|}^{2}}{2\sigma_{n}^{2}}\right\},$$
where $h_{eq,m}^{DF}$ and $h_{eq,m}^{IRS}$ are the *m*-th row vectors of the equivalent channel matrices $H_{eq}^{DF}$ and $H_{eq}^{IRS}$, respectively. The matrix $H_{eq}^{DF}=GS$ represents the channel between the Rx and the DMRUs operating in DF relaying mode, while $H_{eq}^{IRS}=G\left(S⊕I\right)\Phi H$ represents the channel associated with the DMRUs operating in IRS mode.

### 3.2. Design of DMR’s Operating Parameters

To maximize the data rate $R_{DM}$ under the total power constraint $P_{total}$, it is essential to optimize the mode selection matrix S and jointly design the beamforming matrices $W_{Tx}$, $W_{DF}$, and the reflection coefficient matrix Φ. To achieve this object, we first derive the optimal $W_{Tx}$, $W_{DF}$, and Φ that maximize $R_{DM}$ for a fixed S, leading to the optimization problem expressed in Equation (3):(3)$$\begin{array}{cccc} \; & \max_{W_{Tx},W_{DF},\Phi }\; & \; & R_{DM}\\ \; & \mathrm{subject}\;\mathrm{to}\;\left(s.t.\right)\; & \; & \beta_{m}\in \left[0,1\right],\alpha_{m}\in \left[0,2\pi \right),m\in \left\{1,\ldots ,M\right\};\\ \; & \; & & W_{DF}^{H}W_{DF}=I,W_{Tx}^{H}W_{Tx}=I. \end{array}$$

We assume that both $P_{T}$ and $P_{DM}$ are entirely allocated to the data transmission of the target Tx–Rx pair, and that the processing at the Tx and the DF relaying mode DMRUs does not introduce additional gain to the transmitted signal; i.e., $W_{Tx}^{H}W_{Tx}=I$ and $W_{DF}^{H}W_{DF}=I$ hold. Subsequently, for a given total transmit power $P_{total}$, we configure different mode selection matrices S and employ the AO algorithm to solve the optimization problem given in Equation (3). The main idea of the algorithm is as follows. First, initialize Φ=I and design both $W_{Tx}$ and $W_{DF}$ accordingly. Then, based on the determined $W_{Tx}$ and $W_{DF}$, optimize Φ. Subsequently, with the newly obtained Φ, redesign $W_{Tx}$ and $W_{DF}$. This process iterates until the improvement in $R_{DM}$ falls below a predefined threshold, or the number of iterations exceeds a preset maximum value, at which point the algorithm terminates. Thus, for a given S, we can compute a set of $W_{Tx}$, $W_{DF}$, and Φ that maximizes $R_{DM}$. In what follows, we will elaborate on the method for computing the DMR’s operating parameters using AO under a fixed S.

#### 3.2.1. Design of Beamforming at the Tx and DMR

First, we initialize Φ=I (i.e., set $\beta_{m}=1$ and $\alpha_{m}=0$), and apply Zero-Forcing Beamforming (ZFBF) [25] to design $W_{Tx}$ and $W_{DF}$. This method effectively eliminates mutual interference among multiple concurrent signals, thereby improving data transmission performance. Under this initialization, Equation (3) can be rewritten as:(4)$$\begin{array}{cccc} \; & \max_{W_{Tx},W_{DF}}\; & \; & R_{DM}\\ \; & \;\;s.t.\; & \; & \beta_{m}=1,\alpha_{m}=0,m\in \left\{1,\ldots ,M\right\};\\ \; & \; & & W_{DF}^{H}W_{DF}=I,W_{Tx}^{H}W_{Tx}=I. \end{array}$$

We use $v_{DF,m}$ to denote the *m*-th column vector of the matrix $V_{DF}={\left(H_{eq}^{DF}\right)}^{H}{\left[H_{eq}^{DF}{\left(H_{eq}^{DF}\right)}^{H}\right]}^{-1}$, and $v_{Tx,m}$ to represent the *m*-th column vector of $V_{Tx}={\left(H_{eq}^{IRS}\right)}^{H}{\left[H_{eq}^{IRS}{\left(H_{eq}^{IRS}\right)}^{H}\right]}^{-1}$. Here, the inverse exists when the effective channel matrices are full-rank (i.e., all DMRUs operate in the same mode). In hybrid operating modes where the matrices become rank-deficient, the inverse notation ${\left[\cdot \right]}^{-1}$ should be replaced by the Moore-Penrose pseudo-inverse ${\left[\cdot \right]}^{†}$, which provides the minimum-norm solution. Then, the *m*-th column of the precoding matrix $W_{DF}$, denoted as $w_{DF,m}$, can be obtained as:(5)$$w_{DF,m}=v_{DF,m}{∥v_{DF,m}∥}_{F}^{-1},$$
where ${∥\cdot ∥}_{F}$ represents the Frobenius norm of a vector or matrix. Similarly, the *m*-th column of $W_{Tx}$ can be calculated as:(6)$$w_{Tx,m}=v_{Tx,m}{∥v_{Tx,m}∥}_{F}^{-1}.$$

#### 3.2.2. Design of the IRS’s Reflection Coefficient Matrix

After computing $W_{Tx}$ and $W_{DF}$, Equation (4) becomes:(7)$$\begin{array}{cccc} \; & \max_{\Phi }\; & \; & R_{DM}\\ \; & \;s.t.\; & \; & \beta_{m}\in \left[0,1\right],\alpha_{m}\in \left[0,2\pi \right),m\in \left\{1,\ldots ,M\right\}. \end{array}$$

Based on the definition of $s_{m}$ and Equation (2), when $s_{m}=0$, the selection matrix S sets the *m*-th row of $W_{DF}$ to zero, resulting in $h_{eq,m}^{DF}w_{DF,m}=0$. Conversely, when $s_{m}=1$, the operation S⊕I sets the *m*-th main diagonal element of Φ to zero (i.e., $\beta_{m}e^{j\alpha_{m}}=0$), which implies $h_{eq,m}^{IRS}w_{Tx,m}=0$. Substituting $w_{DF,m}=v_{DF,m}{∥v_{DF,m}∥}_{F}^{-1}$ and $w_{Tx,m}=v_{Tx,m}{∥v_{Tx,m}∥}_{F}^{-1}$ into Equation (2) leads to:(8)$$\begin{array}{cc} R_{DM} & =R_{DF}+R_{IRS}\\ \; & \;=\sum_{U_{i}\in \Omega_{DF}}\log_{2}\left\{1+\frac{P_{total}f_{m}^{H}\left(h_{eq,m}^{DF}v_{DF,m}\right){\left(h_{eq,m}^{DF}v_{DF,m}\right)}^{H}f_{m}}{2\sigma_{n}^{2}{∥v_{DF,m}∥}_{F}^{2}}\right\}\\ \; & \;+\sum_{U_{j}\in \Omega_{IRS}}\log_{2}\left\{1+\frac{P_{total}f_{m}^{H}\left(h_{eq,m}^{IRS}v_{Tx,m}\right){\left(h_{eq,m}^{IRS}v_{Tx,m}\right)}^{H}f_{m}}{2\sigma_{n}^{2}{∥v_{Tx,m}∥}_{F}^{2}}\right\}, \end{array}$$
where i,j∈{1,…,M}. Note that the first term on the RHS of Equation (8) does not contain the unknown parameter Φ. Therefore, only the second term on the RHS of Equation (8) needs to be optimized. Consequently, Equation (7) can be equivalently expressed as:(9)$$\begin{array}{cccc} \; & \max_{\Phi }\; & \; & R_{IRS}\\ \; & \;s.t.\; & \; & \beta_{m}\in \left[0,1\right],\alpha_{m}\in \left[0,2\pi \right),m\in \left\{1,\ldots ,M\right\}. \end{array}$$

We can derive:(10)$$h_{eq,m}^{IRS}v_{Tx,m}={\left\{H_{eq}^{IRS}{\left(H_{eq}^{IRS}\right)}^{H}{\left[H_{eq}^{IRS}{\left(H_{eq}^{IRS}\right)}^{H}\right]}^{-1}\right\}}_{m,m}=1,$$
where ${\left(\cdot \right)}_{m,m}$ denotes the element at the *m*-th row and *m*-th column of a matrix. Given that $\left|\right|v_{Tx,m}{\left|\right|}_{F}^{2}=v_{Tx,m}^{H}v_{Tx,m}$ and $v_{Tx,m}^{H}v_{Tx,m}={\left(V_{Tx}^{H}V_{Tx}\right)}_{m,m}$, we can have:(11)$${\left(V_{Tx}^{H}V_{Tx}\right)}_{m,m}=\left\{{\left\{{\left(H_{eq}^{IRS}\right)}^{H}{\left[H_{eq}^{IRS}{\left(H_{eq}^{IRS}\right)}^{H}\right]}^{-1}\right\}}^{H}\right.{\left.\times {\left(H_{eq}^{IRS}\right)}^{H}{\left[H_{eq}^{IRS}{\left(H_{eq}^{IRS}\right)}^{H}\right]}^{-1}\right\}}_{m,m}.$$

Substituting ${\left\{{\left(H_{eq}^{IRS}\right)}^{H}{\left[H_{eq}^{IRS}{\left(H_{eq}^{IRS}\right)}^{H}\right]}^{-1}\right\}}^{H}={\left\{{\left[H_{eq}^{IRS}{\left(H_{eq}^{IRS}\right)}^{H}\right]}^{-1}\right\}}^{H}H_{eq}^{IRS}$ into Equation (11) and simplifying, we obtain:(12)$${\left(V_{Tx}^{H}V_{Tx}\right)}_{m,m}={\left\{{\left[H_{eq}^{IRS}{\left(H_{eq}^{IRS}\right)}^{H}\right]}^{-1}\right\}}_{m,m}^{H}.$$

Since ${\left[H_{eq}^{IRS}{\left(H_{eq}^{IRS}\right)}^{H}\right]}^{-1}$ is a symmetric matrix, the following equation holds:(13)$${\left\{{\left\{{\left[H_{eq}^{IRS}{\left(H_{eq}^{IRS}\right)}^{H}\right]}^{-1}\right\}}^{H}\right\}}_{m,m}={\left\{{\left[H_{eq}^{IRS}{\left(H_{eq}^{IRS}\right)}^{H}\right]}^{-1}\right\}}_{m,m}.$$

Therefore, we get:(14)$$\left|\right|v_{Tx,m}{\left|\right|}_{F}^{2}={\left\{{\left[H_{eq}^{IRS}{\left(H_{eq}^{IRS}\right)}^{H}\right]}^{-1}\right\}}_{m,m}.$$

As a result, $R_{IRS}$ can be simplified as:(15)$$R_{IRS}=\;\sum_{U_{m}\in \Omega_{IRS}}\log_{2}\left\{1+\frac{P_{total}f_{m}^{H}f_{m}}{2\sigma_{n}^{2}{\left\{{\left[H_{eq}^{IRS}{\left(H_{eq}^{IRS}\right)}^{H}\right]}^{-1}\right\}}_{m,m}}\right\}.$$

According to Jensen’s inequality, we have:(16)$$R_{IRS}\leq \log_{2}\left\{1+\;\sum_{U_{m}\in \Omega_{IRS}}\frac{P_{total}f_{m}^{H}f_{m}}{2\sigma_{n}^{2}{\left\{{\left[H_{eq}^{IRS}{\left(H_{eq}^{IRS}\right)}^{H}\right]}^{-1}\right\}}_{m,m}}\right\}.$$

Note that $R_{IRS}$ reaches its maximum value if and only if all non-zero main diagonal elements of ${\left[H_{eq}^{IRS}{\left(H_{eq}^{IRS}\right)}^{H}\right]}^{-1}$ in Equation (16) are equal; this condition can also maximize $R_{DM}$. Suppose that after optimizing Φ, all non-zero main diagonal elements of ${\left[H_{eq}^{IRS}{\left(H_{eq}^{IRS}\right)}^{H}\right]}^{-1}$ become γ, achieving the maximum $R_{DM}$. It is observed that when ${\left[H_{eq}^{IRS}{\left(H_{eq}^{IRS}\right)}^{H}\right]}^{-1}=\gamma \left(S⊕I\right)$, all non-zero main diagonal elements of ${\left[H_{eq}^{IRS}{\left(H_{eq}^{IRS}\right)}^{H}\right]}^{-1}$ are exactly γ; therefore, the condition ${\left[H_{eq}^{IRS}{\left(H_{eq}^{IRS}\right)}^{H}\right]}^{-1}=\gamma \left(S⊕I\right)$ can be used as a constraint for optimizing Φ. Furthermore, using the property of diagonal matrices, we have:(17)$$H_{eq}^{IRS}{\left(H_{eq}^{IRS}\right)}^{H}=1/\left[\gamma \left(S⊕I\right)\right].$$

Thus, to maximize $R_{DM}$, it is necessary to solve for Φ such that the following equation holds:(18)$$H_{eq}^{IRS}{\left(H_{eq}^{IRS}\right)}^{H}=1/\left[\gamma \left(S⊕I\right)\right].$$

The computation procedure is detailed as follows. By expanding ΦH in $H_{eq}^{IRS}=G\left(S⊕I\right)\Phi H$, we can rewrite $H_{eq}^{IRS}$ as:(19)$$H_{eq}^{IRS}=G\left(S⊕I\right)\left[\begin{array}{cccc} h_{1} & \; & & \\ \; & h_{2} & \; & \\ \; & \; & \ddots & \\ \; & \; & & h_{M} \end{array}\right]\left[\begin{array}{c} \phi_{1}\\ \phi_{2}\\ \vdots \\ \phi_{M} \end{array}\right].$$

Here, $h_{m}\in C^{1\times M}$ is the *m*-th row vector of H, and $\phi_{m}=\mathrm{diag}\left(\beta_{m}e^{j\alpha_{m}},\ldots ,\beta_{m}e^{j\alpha_{m}}\right)\in C^{M\times M}$. According to Equation (19), we get:(20)$$H_{eq}^{IRS}{\left(H_{eq}^{IRS}\right)}^{H}=D\left[\begin{array}{c} \phi_{1}\\ \vdots \\ \phi_{M} \end{array}\right]\left[\begin{array}{ccc} \phi_{1}^{H} & \ldots & \phi_{M}^{H} \end{array}\right]D^{H}=\frac{1}{\gamma }{\left(S⊕I\right)}^{-1},$$
where $D\in C^{M\times M}$ is defined as $D=G\left(S⊕I\right)\left[\begin{array}{cccc} h_{1} & & & \\ & h_{2} & & \\ & & \ddots & \\ & & & h_{M} \end{array}\right]$.

Left-multiplying both sides of Equation (20) by the left pseudo-inverse of D, defined as ${\left(D^{H}D\right)}^{-1}D^{H}$, and then right-multiplying by the right pseudo-inverse of $D^{H}$, expressed as $D{\left(D^{H}D\right)}^{-1}$, we obtain:(21)$$\left[\begin{array}{c} \phi_{1}\\ \vdots \\ \phi_{M} \end{array}\right]\left[\begin{array}{ccc} \phi_{1}^{H} & \ldots & \phi_{M}^{H} \end{array}\right]=\frac{1}{\gamma }{\left(D^{H}D\right)}^{-1}\left[D^{H}{\left(S⊕I\right)}^{-1}D\right]{\left(D^{H}D\right)}^{-1}.$$

Expanding the term on the left-hand side (LHS) of Equation (21), and noting that all matrices on the RHS are known without containing any unknown parameters, we can calculate the RHS of Equation (21) and denote the result as $A\in C^{M\times M}$. Thus, we have:(22)$$\left[\begin{array}{ccc} \phi_{1}\phi_{1}^{H} & \ldots & \phi_{1}\phi_{M}^{H}\\ \vdots & \ddots & \vdots \\ \phi_{M}\phi_{1}^{H} & \ldots & \phi_{M}\phi_{M}^{H} \end{array}\right]=\frac{1}{\gamma }A=\frac{1}{\gamma }\left[\begin{array}{ccc} a_{11} & \ldots & a_{1M}\\ \vdots & \ddots & \vdots \\ a_{M1} & \ldots & a_{MM} \end{array}\right],$$
where $a_{pq}\in C^{M\times M}$ (p,q∈{1,…,M} indicate the relative position of $a_{pq}$ within A) is a diagonal matrix whose non-zero main diagonal elements are equal (denoted as $\xi_{pq}$). Since $\beta_{p},\beta_{q}\in \left[0,1\right]$, the main diagonal elements of $\phi_{p}\phi_{q}^{H}$ satisfy $\left|\beta_{p}e^{j\alpha_{p}}\beta_{q}e^{-j\alpha_{q}}\right|\leq 1$. Therefore, the corresponding $\xi_{pq}$ of $a_{pq}$ must also satisfy $\xi_{pq}/\gamma \leq 1$. By defining $\gamma =\sum_{p=1}^{M}\sum_{q=1}^{M}\xi_{pq}$, the condition $\xi_{pq}/\gamma \leq 1$ is satisfied. From Equation (22), we can derive $\phi_{p}\phi_{q}^{H}=a_{pq}/\gamma$, which allows us to establish a set of equations for the unknown variables $\alpha_{1},\alpha_{2},\ldots ,\alpha_{M}$ and $\beta_{1},\beta_{2},\ldots ,\beta_{M}$ as follows:(23)$$\left\{\begin{array}{c} \beta_{m}^{2}=\frac{1}{\gamma }\xi_{mm}\\ \beta_{1}e^{j\alpha_{1}}\beta_{m}e^{-j\alpha_{m}}=\frac{1}{\gamma }\xi_{1m} \end{array}\right.,$$
where m∈{1,…,M}. When p=q, $\xi_{pq}$ is a real number; otherwise, $\xi_{pq}$ is complex. Equation (23) comprises a total of 2M−1 independent equations with 2M unknowns, thus leading to infinitely many solutions. The general solution of Equation (23) for the unknown variables $\beta_{1}e^{\alpha_{1}},\beta_{2}e^{\alpha_{2}},\ldots ,\beta_{M}e^{\alpha_{M}}$ can be expressed as:(24)$$\left\{\begin{array}{c} \beta_{1}e^{j\alpha_{1}}=\sqrt{\frac{\xi_{11}}{\gamma }}e^{j\alpha_{1}}\\ \beta_{m^{\prime }}e^{j\alpha_{m^{\prime }}}=\frac{\sqrt{\gamma }\xi_{1m^{\prime }}}{\sqrt{\xi_{11}}}e^{j\alpha_{1}} \end{array}\right.,$$
where $m^{\prime }\in \left\{2,\ldots ,M\right\}$. To obtain a particular solution, we can assign an arbitrary value to any one of $\beta_{1}e^{\alpha_{1}},\ldots ,\beta_{M}e^{\alpha_{M}}$ (e.g., $\beta_{m}e^{\alpha_{m}}$); afterwards, the values of the remaining $\beta_{m^{\prime }}e^{\alpha_{m^{\prime }}}$ for $m^{\prime }\in \left\{1,\ldots ,M\right\}-\left\{m\right\}$ can be computed using Equation (24).

Subsequently, $W_{TX}$ and $W_{DF}$ are recalculated under the current optimized Φ. Using these updated beamforming matrices, Φ is then re-optimized. This process continues iteratively until the improvement in $R_{DM}$ — defined as the difference between $R_{DM}^{\left(l\right)}$ and $R_{DM}^{\left(l-1\right)}$ obtained in the *l*-th and (l−1)-th iterations — falls below a predefined threshold ε, or until the iteration count *l* reaches the maximum value $l_{max}$. The selection of ε influences the algorithm’s performance and efficiency: a value too large may cause premature termination, resulting in suboptimal data rate performance, while a value too small can yield better optimization at the expense of prolonged convergence. In this work, $\varepsilon =10^{-5}$ is chosen based on simulation studies. To summarize, we present the AO procedure for jointly determining $W_{Tx}$, $W_{DF}$, and Φ in Algorithm 1.

**Algorithm 1** AO-based Joint Optimization of $W_{Tx}$, $W_{DF}$, and Φ
1:**Initialize:** Set iteration counter l=0, max iterations $l_{max}=50$, threshold $\varepsilon =10^{-5}$, and $\Phi_{0}=I$.2:Compute $W_{DF,0}$ and $W_{Tx,0}$ using Equation (3), and calculate $R_{DM,0}$ using Equation (2).3:Update l←l+1.4:Solve for $\Phi_{l}$ based on $W_{DF,l-1}$ and $W_{Tx,l-1}$ using Equation (24).5:Calculate $W_{DF,l}$ and $W_{Tx,l}$ based on $\Phi_{l}$ using Equation (3).6:Calculate $R_{DM,l}$ based on $W_{DF,l}$, $W_{Tx,l}$, and $\Phi_{l}$ using Equation (2).7:If $R_{DM,l}-R_{DM,l-1}>\varepsilon$ and $l\leq l_{max}$, repeat Steps 3–7; otherwise, proceed to Step 8.8:**Output:**$W_{DF}=W_{DF,l}$, $W_{Tx}=W_{Tx,l}$, and $\Phi =\Phi_{l}$. The algorithm ends.


#### 3.2.3. Selection of DMR Operating Mode

For the DMR operating mode selection matrix S, since the DMR contains *M* DMRUs and has a total of $2^{M}$ distinct operating modes, it is necessary to perform joint optimization of $W_{Tx}$, $W_{DF}$, and Φ using the AO technique for each possible S to maximize $R_{DM}$. Then, by comparing $R_{DM}$ across different S configurations, the configuration that yields the highest $R_{DM}$ is selected.

#### 3.2.4. Complexity Analysis of DMRAT

The computational complexity of the proposed DMRAT scheme consists of two parts:

(1) Complexity of the AO algorithm: In each iteration of Algorithm 1, the dominant calculations involve the inversion and multiplication of *M*-dimensional matrices to update the beamforming and reflection matrices. The complexity of these operations is approximately $O(M^{3})$. Assuming an average of $L_{iter}$ iterations for the algorithm to converge, the complexity of the AO algorithm can be expressed as $O(L_{iter}\cdot M^{3})$.

(2) Complexity of mode selection: As described in Section 3.2.3, the DMR comprises *M* units, resulting in a total of $2^{M}$ possible mode combinations. Consequently, the DMRAT scheme needs to perform AO optimization for each candidate mode to maximize the data rate, which incurs additional computational complexity.

Based on the above analysis, the total computational complexity is approximately $O(2^{M}\cdot L_{iter}\cdot M^{3})$. It can be seen that the complexity scales exponentially with *M*, making the practical application of DMRAT challenging. To address the scalability of DMRAT for larger practical systems, machine learning (ML) techniques can be incorporated. For instance, one could train a neural network offline to predict the optimal mode S based on channel inputs, allowing exhaustive search processes to be replaced by low-complexity online inference, significantly reducing the computational burden.

## 4. Evaluation

This section employs MATLAB2025a simulation to evaluate the performance of the DMRAT scheme. The total transmit power is set to $P_{total}=P_{T}+P_{DM}$, with $P_{T}=P_{DM}=P_{total}/2$. Note that when all DMRUs operate in IRS passive reflection mode, $P_{DM}=0$, and accordingly we configure $P_{T}=P_{total}$. We define the transmit power normalized by noise power as $\zeta =10\mathrm{lg}(P_{total}/\sigma_{n}^{2})$ dB, and set ζ∈[0,20] dB for the simulation. We use Monte Carlo simulation with 1500 independent trials. In each trial, the channels H and G are generated randomly.

Figure 2 illustrates the variation in the data rate performance at the Rx along with ζ for different values of *M*. We denote the proposed DMRAT using AO as DMRAT-AO, and compare it with three other schemes: Fixed IRS (where all DMRUs adopt passive IRS reflection mode [24]), Fixed DF relay (where all DMRUs adopt DF relaying mode [26]), and DMRAT-R (where all DMRUs randomly adopt either passive IRS reflection mode or DF relaying mode). As the figure shows, the data rates of all methods increase with the growth of *M*. This is due to the improved signal processing gain provided by multiple antennas. For a fixed *M*, DMRAT-AO outperforms the other three methods in the achievable data rate. This is because DMRAT-AO utilizes AO to jointly optimize the operating parameters of both the Tx and the DMR, selecting the optimal operating mode for the DMR to maximize the data rate at the Rx. The Fixed DF relay scheme employs the DF relaying mode to decode and forward the signal transmitted by the Tx, effectively compensating for channel fading through the consumption of transmit power. Moreover, in this study we assume that the DF relay can decode the signal from the Tx without errors; hence, the rate performance of the Fixed DF relay scheme is influenced solely by the channel between the DMR and the Rx. In other words, under the total transmit power constraint $P_{total}$, the Fixed DF relay scheme allocates half of $P_{total}$ to the Tx, ensuring reliable communication from the Tx to the DMR. In contrast, the Fixed IRS scheme allocates all transmit power to the Tx, and its data rate performance at the Rx is cooperatively affected by the channel fading between the Tx and DMR, as well as the DMR and Rx. Poor quality in either of these two links will degrade the data rate of Fixed IRS. Therefore, given the same ζ, the Fixed DF relay scheme significantly outperforms the Fixed IRS scheme, which exhibits the poorest rate performance. DMRAT-R randomly selects the operating mode of each DMRU. When channel conditions are poor, inappropriate selection of the IRS mode may lead to reduced data rate at the Rx. As a result, the rate performance of DMRAT-R is inferior to that of the Fixed DF relay scheme. To summarize, DMRAT-AO outperforms all other schemes by dynamically exploring a broader solution space that includes all operational modes. By intelligently allocating DMRUs to either passive reflection or active relaying based on channel conditions, DMRAT-AO leverages the advantages of both modes. This approach enables mode selection gain, particularly in scenarios where a single fixed mode is not optimal. As a result, DMRAT-AO maximizes the Rx’s data rate and in dynamic communication environments.

Figure 3 shows the data rate at the Rx versus the number of iterations under M=2 for different DMR operating modes. When all DMRUs operate in DF relaying mode, only ZFBF is required to design $W_{Tx}$ and $W_{DF}$, without needing to compute Φ or apply AO. Therefore, the data rate achieved by the Fixed DF relay scheme remains unchanged as the number of iterations increases. In contrast, the data rate of DMRAT-AO gradually increases during the first five iterations, consistently outperforming the other methods, and converges after approximately five iterations. The data rate of Fixed IRS stabilizes after five iterations, while that of DMRAT-R ceases to improve after four iterations. This is because DMRAT-AO uses AO to select the DMR operating mode that maximizes the data rate at the Rx, resulting in superior performance. Additionally, at the initial iteration (i.e., l=0), since DMRAT-AO sets initial operating state of the DMR to the Fixed DF relay scheme, the data rates of DMRAT-AO and the Fixed DF relay scheme are identical when l=0.

To further evaluate the computational overhead of the proposed DMRAT-AO scheme, we illustrate the average response time in Figure 4 under M∈{2,4,8}. The experiment was conducted on a computer with an Intel Core i5-1240P processor (1.70 GHz), DDR4 16 GB (3200 MHz), and a Windows 11 operating system. As the figure shows, the average response time exhibits a substantial increase with the growth of *M*. This aligns with the complexity analysis in Section 3.2.4. The results indicate that, for small to medium values of *M*, the processing latency of DMRAT-AO remains within an acceptable millisecond range. For larger values of *M*, ML techniques can be incorporated to reduce the latency. Since our primary focus is on the theoretical design of the dual-mode switching mechanism and the potential improvements in data rates enabled by our scheme, the investigation of complexity reduction using ML is left for future study.

The above analysis is based on the assumption of perfect CSI. However, in practical wireless systems, obtaining accurate CSI is often challenging due to factors such as channel estimation errors, quantization effects, and feedback delays. It is therefore crucial to investigate the impact of CSI imperfections on the achievable performance of the proposed scheme. To model the CSI uncertainty, we adopt the widely used additive error model, where the estimated channel matrix $\hat{H}$ is given by [27]:(25)$$\hat{H}=\rho H+\sqrt{1-\rho^{2}}E,$$
where H and $\hat{H}$ denote accurate and inaccurate channel matrices, respectively. The coefficient ρ∈(0,1] indicates the degree of CSI imperfection and ρ=1 means perfect CSI. The matrix E is an $N_{R}\times N_{T}$ complex diagonal Gaussian matrix with zero mean and unit variance, where $N_{R}$ and $N_{T}$ are the numbers of antennas equipped with the receiver and transmitter of a MIMO link.

We applied this model (presented in Equation (Equation 25)) to both the Tx-DMR channel (H) and the DMR-Rx channel (G), and conducted additional simulations under ρ=0.95 [27,28]. Figure 5 illustrates the achievable data rate versus ζ for different methods under M=2, comparing both perfect CSI (ρ=1, solid lines) and imperfect CSI (ρ=0.95, dashed lines) conditions. As Figure 5 shows, all schemes experience performance degradation due to CSI inaccuracies. However, the proposed DMRAT-AO still maintains a significant performance advantage in the imperfect CSI scenario, demonstrating the inherent robustness of its adaptive switching mechanism. Specifically, the Fixed IRS scheme exhibits high sensitivity to CSI errors and suffers from severe performance degradation. This vulnerability arises because its passive beamforming gain relies on precise phase alignment within the cascaded Tx-DMR-Rx channel. Under imperfect CSI, estimation errors from both links become coupled and amplified, resulting in substantial phase mismatches and performance losses. In contrast, the Fixed DF relay scheme demonstrates greater robustness. By utilizing the decode-and-forward mechanism, the DF relaying mode physically decouples the cascaded channel into two independent single-hop links. Although its end-to-end rate is constrained by the bottleneck link (i.e., the minimum rate of the two hops), this decoupling limits the influence of estimation errors to each single hop, effectively preventing the error accumulation and amplification that is inherent to the cascaded IRS link. Regarding the proposed DMRAT-AO, its capability to optimally select the DMR’s operation mode allows it to adaptively switch to the more robust DF mode when CSI inaccuracies severely impair the rate performance of the IRS mode. Consequently, DMRAT-AO can achieve the highest data rate under imperfect CSI compared to the other schemes.

## 5. Conclusions

To address the challenges of high power consumption in active relays and the limited fading resistance of passive IRS, this paper proposes a dual-mode relay (DMR). This relay can dynamically switch between active relaying and passive IRS reflection modes according to channel conditions. Under a total transmit power constraint, we design a DMR-based Adaptive Transmission (DMRAT) method that utilizes AO. DMRAT iterates over all possible DMR operating modes. For each mode, it applies AO to jointly optimize the beamforming matrices at the Tx and the DMR, as well as the IRS reflection coefficient matrix. Consequently, this maximizes the data rate for the target communication pair. The optimal DMR configuration is determined by comparing the maximum achievable rate across all operating modes. The simulation results demonstrate that DMRAT significantly enhances the data rate performance at the intended receiver.

## Figures and Tables

**Figure 1 sensors-25-07492-f001:**
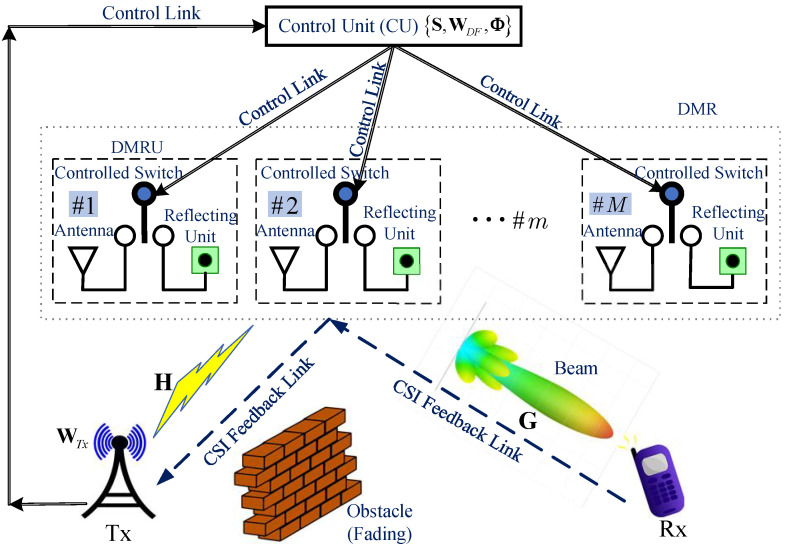
System model.

**Figure 2 sensors-25-07492-f002:**
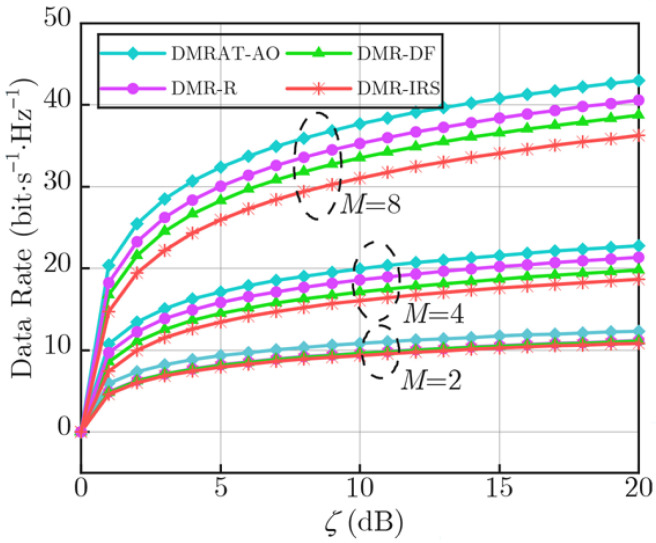
Rx data rate versus ζ for different methods and values of *M*.

**Figure 3 sensors-25-07492-f003:**
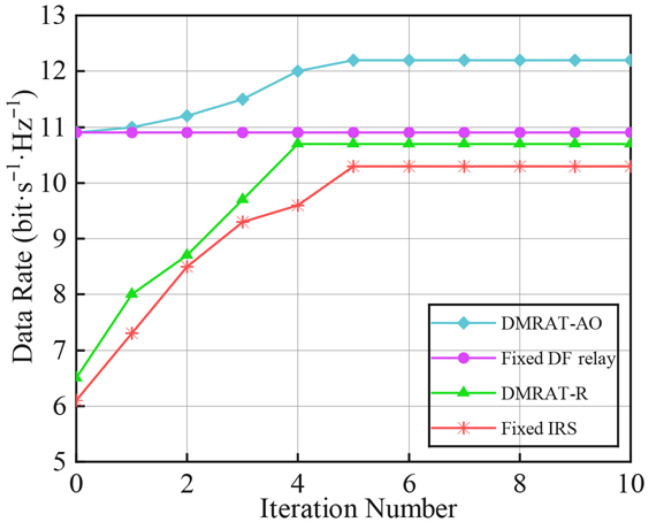
Rx data rate of different methods versus the number of iterations.

**Figure 4 sensors-25-07492-f004:**
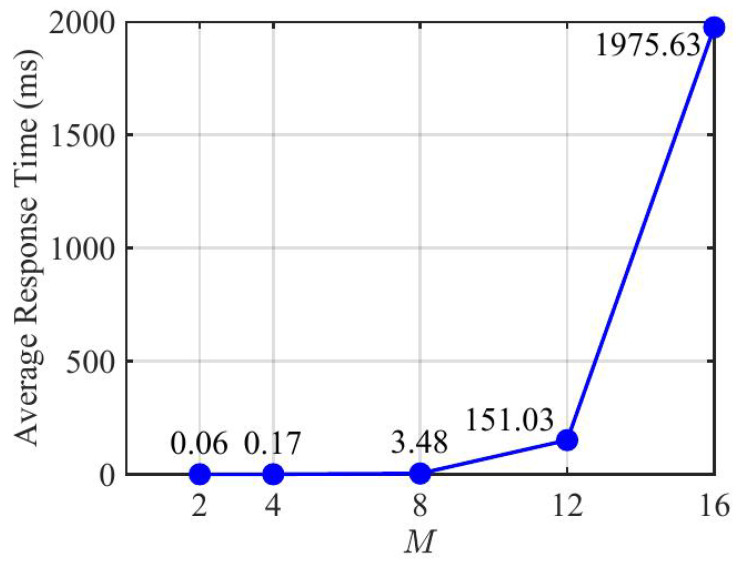
The average response time of DMRAT-AO versus with *M*.

**Figure 5 sensors-25-07492-f005:**
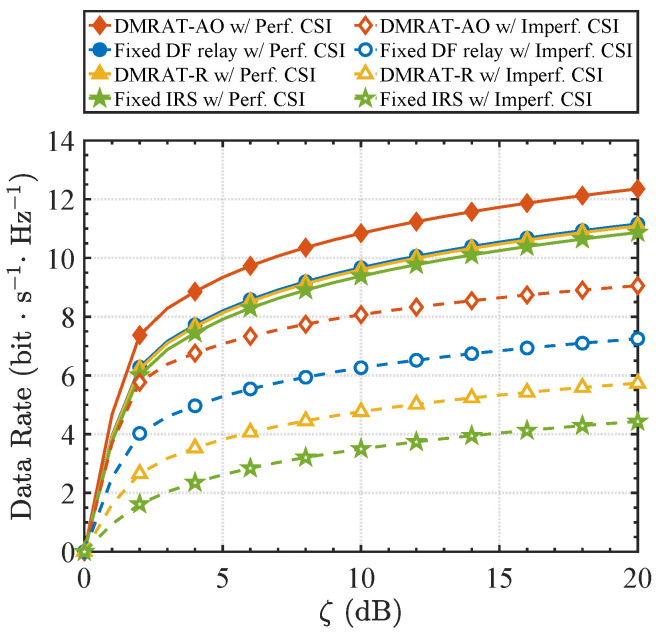
Rx data rate versus ζ for different methods under perfect and imperfect CSI conditions (M=2).

## Data Availability

The data presented in this study are available on request from the corresponding author.

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
