# Peer review of "IRS-Assisted Dual-Mode Relay-Based Adaptive Transmission"

_sensors, 2025, doi:10.3390/s25247492_

Round 1
Reviewer 1 Report
Comments and Suggestions for Authors
The paper proposes a dual mode relay that can switch each DMRU between DF relaying and IRS reflection. For each mode matrix S it uses AO to jointly set WTx, WDF , and Φ, and then it
searches all 2M modes to maximize the data rate in (2). I have the following comments:
- There are missing IRS works in the introduction such as [REF01]. In addition, there are works on RIS-enabled integrated sensing and communication for 6G systems that can be mentioned as well.
- How to obtain all channel state information knowledge ? What is the overhead in terms of pilot structure, identifiability conditions, and the exact number of pilot symbols per coherence block ?
- Please add more information on the caption of figure 1.
- Give the conditions on S and Φ under which (HDFeq(HDFeq)H)-1and (HIRSeq(HIRSeq)H)-1
- exist and are well conditioned.
- The computational complexity of the dual-mode relay (DMR) solution needs to be analyzed.
- Is the convergence of the proposed algorithm converging towards a local or global optimum ?
- More simulations should be conducted to show the effectiveness of the proposed algorithm.
- The insights driven by the receiver data rate versus $\zeta$ for different methods and values of M in figure 2 are not well explained. Please provide the various gains and insights and why proposed approach is attractive.
- The paper has typos like “appear om” which should be “appear on”. Please revise such typos.
References
[REF01] “Multi-Functional RIS for a Multi-Functional System: Integrating Sensing, Communication, and Wireless Power Transfer,” in IEEE Network, doi: 10.1109/MNET.2024.3482571

The English could be improved to more clearly express the research.
Author Response
Dear Editor and Reviewers:
Thank you for the constructive comments from you and the reviewers on our manuscript “IRS Assisted Dual-Mode Relay based Adaptive Transmission”. We have carefully revised the manuscript in compliance with these comments as described below. All of the revised parts are marked in blue in the revised manuscript. We hope the revision and the authors’ reply will be satisfactory to both the reviewers and you. Please let us know if you have any further suggestions/questions regarding our revision and reply. Again, thank you for handling this submission and we look forward to receiving feedback on this revision in the near future.
Best regards,
Dabao Wang, Yanhong Xu, Zhangbo Gao, Hanqing Ding, Shitong Zhu, and Zhao Li

Reviewer 2 Report
Comments and Suggestions for Authors Long sentences are not in the spirit of the English language. Please read and edit the entire paper in this sense, starting with the Abstract. Be sure to define all used variables (see the last paragraph of the Section 2)! All abbreviations must be defined (s.t., ...). Sources must be given for all equations, except for those derived here. This will make the contribution of this paper clearer. Separate the equations which are now incorporated in the in the text in separate rows to be better visible and make the paper more clear! Comments on the Quality of English LanguageLong sentences are not in the spirit of the English language. Please read carefully and improve the whole paper!
Author Response

(The authors gave the same response as above.)

Reviewer 3 Report
Comments and Suggestions for Authors
The overall impression from the submission is quite satisfactory. It is well-written and well-motivated. The authors’ logic is clear and straightforward. The derivations are somewhat easy to follow. Although, there are several important issues that must be addressed.
– In the System Model section, there is an ambiguity between Fig. 1 and its description. In the description, the authors speak about that Rx feeds the sensed CSI to Tx, but there is no such a link in the picture.
– There are some very ambiguous phrases that can be easily assumed to be wrong. For example, the authors state that
“signals actively forwarded and passively reflected by all DMRUs can arrive at the Rx simultaneously,”
or
“signal processing delays for both DF relaying and passive IRS reflection are negligible”,
which is inaccurate due to realistic propagation, processing, and switching delays. Moreover, nowadays, the existing delay is a primary factor limiting their wider application (because of synchronization issues).
– It must be highlighted that the authors presented the assumed system as universally adaptable for wireless environments, without addressing the diversity of real-world topology, mobility, and fading effects. These effects are crucial for most of the systems and can completely ruin the proposed adaptation mechanisms.
– Since an optimization procedure is proposed, a complexity analysis is prerequisite, but it is absent. It is vital to get a better understanding of the real-time operability of such a system.
– Moreover, the authors generalize by saying that "all modes being explored" for maximizing data rate, but is ignores the computational complexity, making the solution impractical for large systems.
– The article implies perfect synchronization between IRS and relay operations; in practice, distributed control introduces significant delays and coordination challenges.
– The application of Zero-Forcing Beamforming for both relay and IRS modes assumes perfect channel state information. Well….this is a natural simplification which is rarely achievable, especially in highly dynamic settings. Thus, it is highly advised to expand the contribution by assuming other types of beamforming. At least in terms of numerical simulation.
– One of the critical issues of the submission is an intolerably small literature review and absolute absence of any comparison with the existing results. There are numerous works that deal with joint analysis of relays, IRS, and MIMO Tx/Rx. So, the contribution is questionable in view of the absence of any comparison with the existing results.
Author Response

(The authors gave the same response as above.)

Round 2
Reviewer 1 Report
Comments and Suggestions for Authors
Thank you for addressing most my comments. I have the remaining minor comments left:
- Please double check reference formats.
- You can also add more works on IRS such as [REF01] in addition to IRS applications for low dynamic range for ris-aided bistatic systems.
References
[REF01] A. Bazzi and M. Chafii, "Towards ISAC RIS-Enabled Passive Radar Target Localization," ICC 2025 - IEEE International Conference on Communications, Montreal, QC, Canada, 2025, pp. 2290-2295, doi: 10.1109/ICC52391.2025.11162110.
Reviewer 2 Report
Comments and Suggestions for Authors
I see no reason to include a footer when that text can be written within the main text.
Also, the title of Figure 1 is sufficient and the rest of the text below it can be included in the description of that figure in the text of the paper.
Some abbreviations still are not defined when used for the first time (UAV, AWGN, …)! Check the whole paper once again!
Reviewer 3 Report
Comments and Suggestions for Authors
The authors performed a careful revision, clearing up most of the issues raised. I see no further obstacles to publication of the submission.
